# Langevin Dynamics Study on the Driven Translocation of Polymer Chains with a Hairpin Structure

**DOI:** 10.3390/molecules29174042

**Published:** 2024-08-26

**Authors:** Fan Wu, Xiao Yang, Chao Wang, Bin Zhao, Meng-Bo Luo

**Affiliations:** 1Department of Physics, Taizhou University, Taizhou 318000, China; yangxiao@tzc.edu.cn (X.Y.); chaowang0606@126.com (C.W.); zhaobin007@tzc.edu.cn (B.Z.); 2Department of Physics, Zhejiang University, Hangzhou 310027, China

**Keywords:** hairpin, polymer, translocation, nanopore, Langevin dynamics simulation

## Abstract

The hairpin structure is a common and fundamental secondary structure in macromolecules. In this work, the process of the translocation of a model polymer chain with a hairpin structure is studied using Langevin dynamics simulations. The simulation results show that the dynamics of hairpin polymer translocation through a nanopore are influenced by the hairpin structure. Hairpin polymers can be classified into three categories, namely, linear-like, unsteady hairpin, and steady hairpin, according to the interaction with the stem structure. The translocation behavior of linear-like polymers is similar to that of a linear polymer chain. The time taken for the translocation of unsteady hairpin polymers is longer than that for a linear chain because it takes a long time to unfold the hairpin structure, and this time increases with stem interaction and decreases with the driving force. The translocation of steady hairpin polymers is distinct, especially under a weak driving force; the difficulty of unfolding the hairpin structure leads to a low translocation probability and a short translocation time. The translocation behavior of hairpin polymers can be explained by the theory of the free-energy landscape.

## 1. Introduction

The geometric structures of polynucleotides such as DNA and RNA are often complex. The hairpin structure is one such complex geometric structure. The simplest hairpin (or stem-loop) structure essentially consists of a double-stranded (ds) stem and a single-stranded (ss) loop, and it can be found in almost every prediction of polynucleotide folding [1]. It is well known that the translocation behavior of a polymer chain through nanopores is strongly related to its structure [2]. Therefore, studies on the translocation of polymer chains with hairpin structures can give insights into the structural characteristics of the hairpin, which are related to biological processes such as DNA replication, recombination, and repair, and RNA transcription and interference [3,4,5,6].

Nanopore force spectroscopy (NFS) is an ideal single-molecular tool for studying the kinetics of linear polymers such as ssDNA and has been widely used over the last two decades [7,8,9,10,11,12,13]. A polymer driven through a nanopore by a transmembrane electrical field could significantly reduce the ion current across the nanopore. The dynamics of the polymer can be detected from the ion current profile. Experiments have shown that, for a structured polymer such as hairpin RNA with a stem whose size is larger than the nanopore [1,14,15], the translocation process cannot be completed unless the driving force is strong enough to unfold the stem [16,17,18]. The minimum driving force required to unfold the stem is called the critical unfolding driving force. Ion current profiles have also shown that the time interval required for the unfolding of hairpin structures is much longer than that for the translocation of a linear chain with the same contour length [19,20,21,22,23,24].

In addition to experimental measurements, theoretical calculations and simulations enable us to quickly and quantitatively understand the mechanisms of polymer translocation and obtain information about the general laws behind various associated phenomena. For instance, Bockelmann et al. found, through theoretical calculations, that the unfolding process of a hairpin structure can be treated as the thermal passage of a single energy barrier whose landscape is determined by the sequential base pair opening [25,26]. At present, there are relatively few all-atom molecular dynamics simulation studies on the translocation of hairpin polymers through nanopores. This is because the time taken for the translocation (millisecond level [2]) of hairpin polymers through a nanopore is much longer than the current simulation time span (nanosecond level) applicable to common molecular dynamics methods. Therefore, all-atom molecular dynamics simulations are only suitable for short hairpin polymers under large driving forces. For example, Comer et al. used the molecular dynamics of the Amber model [27] and Stachiewicz et al. used the improved Martini model and Brownian dynamics [28,29] to successfully simulate the translocation of hairpin polymers with 12 and 10 base pairs through nanopores. Although the translocation times (30 ns and 70 ns, respectively) fell within a valid time span for computer simulation, the driving force corresponding to the experiment required a large driving voltage to achieve such fast unfolding, which was far beyond the tolerance range of the nanopore. Thus, simulations on longer hairpin polymer chains with a smaller driving voltage are needed.

A commonly used simulation method in studying the translocation of linear polymers through nanopores is the Langevin dynamics (LD) simulation method using the coarse-grained bead–spring (CGBS) model [30,31,32,33,34]. Using the CGBS model, ssDNA chains are modeled as a series of beads connected by a spring, and the ions in the solvent are modeled as an average field. These treatments can considerably reduce the degrees of freedom of the system and improve the simulation efficiency, so that one can simulate a complex dynamic process with a long time span. For instance, Muthukumar et al. and Luo et al. applied LD simulations to the translocation processes of ssDNA [30,31,32], dsDNA [33], and star-shaped macromolecules [34]. It was found that the simulation results were highly consistent with the experimental results. 

In this work, the influence of the hairpin structure on the translocation process is studied using the Langevin dynamics simulation method and the Fokker–Planck theory. The simulation results are consistent with the theoretical results, which can be explained by the theory of the free-energy landscape during the translocation. This section is followed by the introduction of the simulation and theoretical calculation methods, the results and discussion, and the conclusions.

## 2. Simulation and Theoretical Calculation

### 2.1. Simulation

The translocation of hairpin polymers through nanopores is simulated in a three-dimensional (3D) system. The 3D system is presented in Figure 1a as a 2D sketch. The hairpin polymer is modeled using the CGBS model. *N* beads with diameter σ are connected into a linear chain and then folded into the hairpin structure, with the hairpin structure consisting of a single-stranded overhang part, a double-stranded stem part, and a single-stranded loop. The lengths of each part are set as N_hang_, *N*_stem_, and *N*_loop_, respectively; thus, the chain length *N* = *N*_hang_ + 2*N*_stem_ + *N*_loop_. As shown in Figure 1b, each monomer is marked sequentially with the serial number *i* = 1 to *N* according to its position along the chain. The monomers in the first stem part (*N*_hang_ + 1 ≤ *i* ≤ *N*_hang_ + *N*_stem_) are sequentially bonded to the monomers in the second stem part (*N* ≥ *i* ≥ *N* − *N*_stem_ + 1) through hydrogen bond (HB) interactions, thus forming base pairs (bps) with serial number *i*_bp_ = 1 to *N*_stem_.

The size of the system in the *x* direction is much larger than the contour length of the polymer chain, and periodical boundary conditions (PBCs) are considered in the *y* and *z* directions, so the size effect of the system on the translocation can be neglected. An impenetrable wall is set parallel to the *yz* direction at position *x* = 0. The wall consists of a single layer of beads with diameter σ. The wall beads are motionless during the simulation, and this divides this system into a *cis* side (*x* < −σ/2) and a *trans* side (*x* > σ/2). The bead at the center of the wall is dug out to form a square nanopore with side length σ, which allows single strands to pass sequentially while blocking double strands.

In the LD simulation, the positions of the monomers in the system are determined by their coordinates, which are updated by solving the Langevin equation:(1)$$m\frac{d^{2}{\vec{r}}_{i}}{dt^{2}}=-\nabla U+{\vec{F}}^{\left(T\right)}-\eta {\vec{v}}_{i}+\vec{f},$$
where ${\vec{r}}_{i}$ is the position of the *i*th monomer, and *U* is the potential energy of the polymer monomer, which consists of the modified finitely extensible nonlinear elastic (FENE) potential for bonded monomers *U*_FENE_, the truncated and shifted Lenard–Jones (LJ) potential for non-bonded monomers *U*_LJ_, the screened Coulomb potential for charged monomers *U*_Coul_, the HB interaction for monomers in the stem part *U*_HB_, and the purely repulsive Weeks–Chandler–Andersen (WCA) potential between the chain monomer and wall bead *U*_WCA_ (*U* = *U*_FENE_ + *U*_LJ_ + *U*_Coul_ + *U*_HB_ + *U*_WCA_). Details of these interactions (except for the HB potential) can be found in our previous simulation study on the translocation of a semi-flexible polymer through a nanopore [35]. The HB potential *U*_HB_ in the stem part can be represented by the LJ potential [36]:(2)$$U_{\mathrm{HB}}(r_{\mathrm{stem}})=\left\{\begin{array}{ll} 4\varepsilon_{\mathrm{stem}}\left[{\left(\frac{\sigma }{r_{\mathrm{stem}}}\right)}^{12}-{\left(\frac{\sigma }{r_{\mathrm{stem}}}\right)}^{6}\right]-\;4\varepsilon_{\mathrm{stem}}\left[{\left(\frac{\sigma }{r_{\mathrm{stem}\_\mathrm{cut}}}\right)}^{12}-{\left(\frac{\sigma }{r_{\mathrm{stem}\_\mathrm{cut}}}\right)}^{6}\right] & ,\;r_{\mathrm{stem}}<r_{\mathrm{stem}\_\mathrm{cut}}\\ \;0\; & ,\;r_{\mathrm{stem}}\geq r_{\mathrm{stem}\_\mathrm{cut}} \end{array}\right.,$$
where *r*_stem_ is the distance between two monomers in one base pair (bp); *ε*_stem_ is the interaction strength for the HB potential; *r*_stem_cut_ is the cutoff distance for the HB potential, set as 2.5 for attractive interaction (small changes in the value of *r*_stem_cut_ barely influence the simulation results); ${\vec{F}}^{\left(T\right)}$ is the random thermal force with mean *<F>* = 0 and variation <*F^2^*> = 6*ηk*_B_*T*, where *k*_B_ is the Boltzmann constant and *T* is the absolute temperature; $-\eta {\vec{v}}_{i}$ is the viscous force, which is proportional to the viscosity coefficient *η* and the speed of the *i*th bead ${\vec{v}}_{i}$; and $\vec{f}$ is the electrical driving force, which is proportional to the electrical field $\vec{E}$ ($\vec{f}=q\vec{E}$).

The position and velocity $\left\{{\vec{r}}_{i},{\vec{v}}_{i}\right\}$ of the *i*th monomer in the polymer are updated via the modified velocity Verlet algorithm:(3)$$\begin{array}{l} {\vec{r}}_{i}(t+\Delta t)={\vec{r}}_{i}(t)+\Delta t{\vec{v}}_{i}(t)+\frac{1}{2}\Delta t^{2}{\vec{a}}_{i}(t)\\ \vec{v}\prime_{i}(t+\Delta t)={\vec{v}}_{i}(t)+\lambda \Delta t{\vec{a}}_{i}(t)\\ {\vec{a}}_{i}(t+\Delta t)=\vec{a}({\vec{r}}_{i}(t+\Delta t),\vec{v}{’}_{i}(t+\Delta t))\\ {\vec{v}}_{i}(t+\Delta t)={\vec{v}}_{i}(t)+\frac{1}{2}\Delta t[{\vec{a}}_{i}(t)+{\vec{a}}_{i}(t+\Delta t)] \end{array},$$
where *m*, ${\vec{r}}_{i}$, ${\vec{v}}_{i}$, and ${\vec{a}}_{i}$ are, respectively, the mass, position, velocity, and acceleration of the *i*th monomer, and the parameter *λ* is set to 0.6 in the modified velocity Verlet algorithm.

At the initiation of the simulation, a linear polymer chain is generated at the *cis* side near the nanopore. Then, the linear polymer is diffused for a sufficiently long time with the head monomer (marked by the serial number 1) at the entrance of the nanopore. Once the polymer reaches the equilibrium state, adding the HB interaction to the stem part and setting the simulation time as *t* = 0, the linear polymer is folded into the hairpin structure under the effect of the HB interaction, and the time interval for this period is determined as the folding time for the hairpin structure. Then, the hairpin polymer is diffused for another long period of time with the head monomer fixed near the nanopore until the chain reaches the equilibrium state again. The simulation time is reset to *t* = 0 before the chain is driven to translocate through the nanopore. In the simulation, the unit of time is $\tau_{\mathrm{LJ}}=\sqrt{\frac{m\sigma^{2}}{k_{B}T}}$. During translocation, the monomers are translocated through the nanopore sequentially according to their serial number. Once *m* monomers are translocated into the *trans* side, the monomer with serial number *m* + 1 is located in the nanopore, so that the number of monomers remaining on the *cis* side is *N* − *m* − 1. The translocation state of the polymer can be determined from the number of translocated monomers *m* on the *trans* side; *m* increases sequentially from 0 to *N*, and the translocation is completed when *m* = *N*.

Since the absorbing–absorbing boundary conditions are adopted in this simulation, the polymer chain may be drawn back to the *cis* space even if several monomers have entered the *trans* side due to the random thermal motion, thus contributing to an attempted translocation. The time interval for this period is called the attempted translocation time. The simulation time is reset to 0 if the chain threads back into the *cis* side, and the polymer chain attempts to translocate through the nanopore for a second time. After several failed attempts (the number of times is noted as *N*_trail_), successful translocation can be observed with all monomers entering into the *trans* space, and the time interval for this final successful translocation is defined as the translocation time *τ*. The translocation probability *P*_trans_ is defined as the ratio of the number of successful translocation attempts to the total number of translocation attempts, i.e., *P*_trans_ = 1/(1 + *N*_trial_).

Due to the hydrogen bond interaction, the monomers in the stem part may be trapped near the nanopore for a long time and undergo a sub-diffusion process during translocation. The sub-diffusion of these monomers will have a certain impact on the accuracy of the LD simulation results. The physical mechanism behind sub-diffusion is quite complex, and it is challenging to identify appropriate simulation methods and parameters. Therefore, we can only minimize the impact of sub-diffusion on the entire translocation process by controlling the length of and interaction strength of the stem part.

### 2.2. Theoretical Calculation

In the theoretical calculation of hairpin polymer translocation, the number of translocated monomers *m* increases one by one from 0 to *N*; at each time step, *m* increases by one. Assuming that the translocation of hairpin polymers is slow, the polymer chain is in an equilibrium state during the translocation process, so the free energy *F*(*m*) for a hairpin polymer can be calculated for each time step. The free energy is expressed as *F* = *U* − *TS*, where *U* and *S* represent the potential energy of the polymer chain and the polymer configuration’s entropy, respectively. Since the wall is very thin, the polymer chain during translocation can be considered equivalent to two end-grafted polymer chains. The polymer on the *trans* side is a linear chain with length *m* (assuming that the polymer on the *trans* side no longer folded into the hairpin structure), while the polymer on the *cis* side is a hairpin polymer with length *N* − *m*. If the unit of energy in the calculation is set as *k*_B_T, the entropy part *TS* can be expressed as TS=(1−γ)ln[m(N−m)]+Nlnμ [35]. Here, *γ* and μ represent the scaling exponent and the effective coordination number of the configuration number for an end-grafted polymer of *m* monomers, respectively; *γ* = 0.69 is applied in this calculation for the three-dimensional self-avoiding walk chain. The potential *U* is related to the folding energy of the stem part, which is assumed to be proportional to the interaction and the length of the stem part −*e*_stem_*N*_stem_; the coefficient *e*_stem_ is the folding energy for one base pair in the theoretical calculation. Thus, the free-energy landscape of the polymer F(*m*) can be expressed as
(4)$$F(m)=\left\{\begin{array}{ll} (1-\gamma )\ln N_{hang}-mqV-e_{stem}N_{stem} & m=0,N_{hang}\\ (1-\gamma )\ln[m(N_{hang}-m)]-mqV-e_{stem}N_{stem} & 0<m<N_{hang}\\ \left(1-\gamma )\ln[m(m-N_{hang})]-mqV-e_{stem}(N_{stem}+N_{hang}-m\right) & N_{hang}<m<N_{hang}+N_{stem}\\ (1-\gamma )\ln[m(N-m)]-mqV & N_{hang}+N_{stem}\leq m<N\\ (1-\gamma )\ln N-mqV & m=N \end{array}\right.$$
where the last term −*mqV* represents the electrical driving force; the magnitude of charge *q* is set as *q* = 1; and *V* is the electrical potential difference between the *cis* and *trans* sides, so that the driving force *f* = *V*.

Considering that the translocation process for hairpin polymers is very slow and close to the equilibrium state, the translocation of hairpin polymers can be treated as a diffusive random process, which can be described via a Fokker–Planck equation [8]
(5)$$\frac{\partial }{\partial t}p(m,t)=\frac{1}{b^{2}}\frac{\partial }{\partial m}D(m)e^{-F(m)}\frac{\partial }{\partial m}e^{F(m)}p(m,t),$$
where *p*(*m*, *t*) is the probability distribution and *D*(*m*) is the diffusion coefficient. Considering the coarse-grained model applied in the simulation, the monomers are homogeneous, with the exception of the HB interaction, which has been considered in the calculation of the free-energy landscape. Thus, the efficient *D* can be assumed to be constant and independent of *m* in the calculation; *b* = 1 is the mean bond length; and *b*^2^/*D* is taken as the unit of time in the calculation. 

The translocation time *τ*_FP_ can be calculated by solving the Fokker–Planck equation with absorbing–absorbing boundary conditions [37]:(6)$$\tau_{\mathrm{FP}}=\frac{\Psi (0,\;1)\phi (1,\;N)-\Psi (1,\;N)\phi (0,\;1)}{\Psi (0,\;1)\Psi (0,\;N)}.$$

Here, *Ψ*(*n*_1_, *n*_2_) and *ϕ*(*n*_1_, *n*_2_) capture the information about the free-energy landscape and can be calculated as $\Psi (n_{1},n_{2})=\int_{n_{1}}^{n_{2}}\frac{1}{\psi (n)}dn$ with $\psi (n)=\exp\left[-\int_{1}^{n}\frac{\partial F_{n’}}{\partial n’}dn’\right]$ and $\phi (n_{1},n_{2})=\int_{n_{1}}^{n_{2}}\frac{1}{\psi (n)}dn\int_{1}^{n}\frac{\psi (n^{\prime })\Psi (0,n^{\prime })}{u_{0}}dn^{\prime }$, respectively.

## 3. Results and Discussion

### 3.1. Hairpin Structure

We first examine the conditions for the formation of a stem in the polymer chain. The lengths of the polymer chain and each part are set as *N* = 45, *N*_hang_ = 30, *N*_stem_ = 5, and *N*_loop_ = 5, respectively, and the interaction strength for the HB potential is set as *ε*_stem_ = 6. The lower part of Figure 2 shows the relationship between the average distance of each base pair in the stem part <*r*_stem_> and the simulation time *t*; different lines represent base pairs in different positions (*i*_bp_ = 1 ~ 5). We find that <*r*_stem_> decreases with *t* and then tends to be stable for all *i*_bp_. The base pairs near the chain end (*i*_bp_ = 1) decrease slowly and the final <*r*_stem_> is large, while the base pairs near the loop (*i*_bp_ = 5) decrease rapidly and the final <*r*_stem_> is small, being larger than 1 considering the excluded volume effect. If we consider that the hydrogen bond is formed if the distance <*r*_stem_> is less than 2, the number of hydrogen bonds <*N*_bp_> increases with the simulation time *t*. The upper part of Figure 2 shows the relationship between <*N*_bp_> and simulation time *t* for polymers with different *ε*_stem_. <*N*_bp_> increases with *t* and then tends to be stable, while, when the interaction *ε*_stem_ is small, <*N*_bp_> is less than 5, even if the simulation time is long enough. We find from the dashed line that the value of <*N*_bp_> for a linear polymer when *t* = 100 is not 0 but a small number less than 1. This is attributed to the standard adopted when judging base pair formation via the distance, since the distance among the monomers in a linear chain can be very short, even if there is no HB interaction. To verify this, the critical distance for base pair formation is decreased from 2 to 1.5, and the value of <*N*_bp_> for a linear polymer when *t* = 100 decreases (but is still not 0), while <*N*_bp_> for a hairpin polymer is still close to 5.

The dependence of the base pair ratio <*N*_bp_/*N*_stem_> when *t* = 100 on the interaction *ε*_stem_ for polymers with different numbers of base pairs *N*_stem_ is plotted in Figure 3. We find that the ratio <*N*_bp_/*N*_stem_> is increased gradually with *ε*_stem_ and gradually approaches 1 at a large *ε*_stem_. We also find that, for polymers with only one or two base pairs, the increase in <*N*_bp_/*N*_stem_> is slower, meaning that the polymer requires greater interaction strength to form a hairpin structure. For the case with *N*_stem_ ≥ 3, <*N*_bp_/*N*_stem_> reaches 1 at approximately *ε*_stem_ = 4. The inset of Figure 3 shows the variation rate d<*N*_bp_/*N*_stem_>/d*ε*_stem_. We can see that, for polymers with *N*_stem_ ≥ 3, the maximum point *ε*_stem_* for the changing rate is approximately *ε*_stem_* = 2, and the ratio <*N*_bp_/*N*_stem_> is approximately 0.5 when *ε*_stem_ = 2 from the main plot. Thus, for *N*_stem_ ≥ 3, we can divide the attractive strength into three regions: a linear polymer region when *ε*_stem_ < 2; an unsteady hairpin polymer region at 2 < *ε*_stem_ < 4; and a steady hairpin polymer region at *ε*_stem_ > 4.

### 3.2. Translocation Process of Hairpin Polymer

We have studied the influence of the attraction strength of the stem part *ε*_stem_ on the translocation process. Figure 4 shows the dependence of the average translocation time <*τ*> (main plot) and the average translocation probability <*P*_trans_> (inset) on *ε*_stem_. The lengths of each part of the hairpin polymer chain are *N*_hang_ = 30, *N*_loop_ = 5, and *N*_stem_ = 5, respectively. The driving force is set as *f* = 0.2 and *f* = 0.3, respectively. The translocation time for *ε*_stem_ = 0 is noted as <*τ*_0_> for the linear polymer chain. According to the behavior of the mean translocation time <*τ*> and the average translocation probability <*P*_trans_>, the attraction strength *ε*_stem_ can be divided into three regions. (1) When *ε*_stem_ is less than *ε*_stem_* = 2, there is no steady stem structure in the polymer chain. The translocation process here is similar to that of a linear chain. We can see from the plot that <*τ*> is close to <*τ*_0_> and the probability <*P*_trans_> is the same as that of a linear chain in this region. (2) When *ε*_stem_ is larger than 2 but less than approximately 4, an unsteady hairpin structure is formed in this region. It takes a long time to break the bonded base pair, and a larger *ε*_stem_ results in a longer time, so that the translocation time <*τ*> increases gradually with the interaction *ε*_stem_. Meanwhile, as the hairpin structure is not steady, it can eventually be broken, and the probability <*P*_trans_> is still the same as that of a linear chain in this region. (3) When *ε*_stem_ is larger than 4, the translocation time <*τ*> approaches a stable value, while the probability <*P*_trans_> decreases with *ε*_stem_. This is because the hairpin structure in the polymer is very stable in this region. Bonded base pairs cannot be broken easily, and, considering the absorbing–absorbing boundary condition applied in this simulation, the chain is more inclined to thread back into the *cis* space, so the number of trial times *N*_trial_ increases and the probability <*P*_trans_> decreases with *ε*_stem_. The reason that the translocation time <*τ*> tends to be stable will be explained in detail in the following part. Note that the critical values for *ε*_stem_ when distinguishing between the three regions are not constant and are also influenced by the driving force *f*. This will also be discussed in detail later.

To obtain more information about the translocation time, we have calculated the mean residence time for each monomer remaining inside the nanopore during translocation. The translocation state can be determined via the number of translocated monomers *m* on the *trans* side. During translocation, the monomers with serial number *m* + 1 pass through the nanopore in turn. The residence time of the (*m* + 1)th monomer is noted as <*t_m_*_+1_>, and the translocation time is the sum of the residence times of all monomers in the chain, i.e., $<\tau >=\sum_{m=0}^{N-1}<t_{m+1}>$. The distribution of <*t*_m+1_> with *m* for polymers with different attraction strengths *ε*_stem_ is shown in Figure 5. We find that the solid lines denoting <*t*_m+1_> for hairpin polymers are similar to the dashed lines denoting the linear chain, because the overhang part and the loop part of the hairpin polymer exhibit a single-stranded linear chain. As the breakage of the base pairs in the stem part takes a long time, the <*t*_m+1_> for the stem part is larger than that of the linear chain; in particular, it reaches a peak at the point *m* = *N*_hang_ − 1, forming a triangle enclosed by the solid and dashed lines in the range of *m** < *m* < *N*_hang_ + *N*_stem_. The difference in the translocation time between the hairpin and linear polymers <*τ*> − <*τ*_0_> is mainly dependent on the area of the triangle, which is related to the interaction *ε*_stem_. When *ε*_stem_ = 1, the hairpin chain is close to the linear chain and the area is very small, so that <*τ*> is close to <*τ*_0_>. Meanwhile, for an unsteady hairpin polymer, <*t*_m+1_> increases and *m** decreases as *ε*_stem_ increases, so that the area and <*τ*> are increased accordingly. Moreover, for a steady hairpin polymer, when *ε*_stem_ is greater than 4, *m** decreases to 0, meaning that the whole chain threads back to the *cis* side, leading to a decrease in the translocation probability <*P*_trans_>. Meanwhile, <*t*_m+1_> no longer increases, so that the area and <*τ*> no longer increase. We also find that, for a steady hairpin polymer, the <*t*_m+1_> for the monomers in the loop part is slightly lower than the dashed line. A reasonable assumption is that these monomers are packed near the nanopore due to the loop structure, so that they can easily be dragged into the nanopore by the monomers in the stem part with high free energy, which will be discussed in detail later.

To further study the residence time, we theoretically calculate the free energy *F*(*m*) and the translocation time *τ*_FP_ of the hairpin polymers with different stem interactions *e*_stem_ during translocation, and the results are shown in Figure 6. We can see from the plot (d) that although the theoretical parameters of *τ*_FP_ and *e*_stem_ are not numerically identical to *τ* and *ε*_stem_ in the simulation, the dependence of *τ*_FP_ on *e*_stem_ is consistent with the simulation results in Figure 4, which also shows three regions. The different free-energy landscapes for the three regions are shown in plots (a) to (c), respectively. The dashed line (linear polymer) decreases monotonically under the effect of the driving force, while the sold line (hairpin polymer) firstly decreases from a lower starting point, reaching a minimum at *m* = *N*_hang_ − 1, and then increases with the unfolding of the stem structure until it coincides with the dashed line when *m* > *N*_hang_ + *N*_stem_ [25,26]. The non-monotonic change in the solid line leads to an energy well in the free-energy landscape of the hairpin polymer. The position of the energy well is coincident with the range of the triangle area in Figure 5. *m** is defined as the point where *F*(*m**) = *F*(*N*_hang_ + *N*_stem_), and the residence time can be explained by the position of the monomer trapped in the well. The peak point of <*t*_m+1_> corresponds to the bottom of the energy well, and <*t*_m+1_> decreases as the monomers move away from the bottom. The monomers in the well can either overcome the folding energy barrier *H*_right_ to pass into the *trans* side or overcome the left barrier *H*_left_ to thread back into the *cis* side, so that the translocation behavior is related to the width and depth of the free-energy well.

The value of *H*_left_ is related to the driving force *f* and the length of the overhang part *N*_hang_, and *H*_right_ is dependent on the coefficient *e*_stem_ and the length of the stem part *N*_stem_. *H*_left_ is a constant when *f*, *N*_hang_, and *N*_stem_ are all fixed, while *H*_right_ increases with *e*_stem_. As shown in plot (a), when *e*_stem_ is small, *H*_left_ >> *H*_right_, and the depth and width of the energy well are both small, the monomers can easily pass through the nanopore, so the translocation time changes slightly with *e*_stem_, corresponding to the first region in Figure 4. Meanwhile, as shown in plot (b), as *e*_stem_ increases, the depth of the well increases with *H*_right_ until it is equal to *H*_left_, and the width of the well increases as *m** decreases to 0. The monomers still tend to translocate onto the *trans* side, but a longer translocation time is needed, corresponding to the rapidly increasing region of <*τ*> in Figure 4. When *e*_stem_ is very large, *H*_right_ > *H*_left_, the depth (*H*_left_) and width (*m** = 0) of the energy well exhibit a larger change, so that the translocation time no longer changes in this region. Moreover, in this region, the whole chain can thread back into the *cis* side, and an increase in *H*_right_ indicates an increase in the number of failed attempts, which leads to a decrease in the translocation possibility *P*_trans_.

From the free-energy landscape, if the length of the overhang part *N*_hang_ is changed, the location of the energy well and its depth are changed accordingly; thus, we study the influence of *N*_hang_ on the translocation time *τ*, and the simulation results are shown in Figure 7. The theoretical results, which are not plotted in the figure, show the same trend. We can see that as *N*_hang_ increases, *τ* initially increases with *N*_hang_ and then reaches a stable value. This can be clearly determined from the free-energy landscape shown in the inset. When *N*_hang_ is small (left inset), *H*_left_ < *H*_right_, the depth of the well (*H*_left_) increases as *N*_hang_ increases, causing an increase in the translocation time *τ*. When *N*_hang_ is large enough (right inset), *H*_left_ > *H*_right_, the depth of the well (*H*_right_) no longer changes with *N*_hang_; at this time, the translocation time *τ* will not change.

We also see in Figure 4, Figure 5 and Figure 7 that the lines for the translocation times under different driving forces are crossed. For instance, in Figure 7, when *N*_hang_ is small, the translocation time *τ* for *f* = 0.2 is smaller than that for *f* = 0.3. Meanwhile, when *N*_hang_ is large, *τ* for *f* = 0.2 is larger than that for *f* = 0.3, indicating that the driving force does not simply facilitate the translocation. The influence of the driving force *f* on the translocation time is studied using the simulation method, and the results are shown in Figure 8. The dependence of the translocation probability on *f* is also plotted in the inset of Figure 8. We can see that for a linear polymer (dashed line), the translocation time <*τ*_0_> decreases as the driving force *f* increases via the power law <*τ*_0_> ~ *f*
^− *δ*^. The exponent *δ* = 0.92, and the value of *δ* is less than 1 because the polymer chain is not in an equilibrium state during translocation. *δ* is slightly larger than the value of 0.8 that we obtained in an earlier work. This is because the size of the nanopore applied in this work is smaller, causing a very slow translocation process that is closer to equilibrium. The reduction in the nanopore size also leads to a lower translocation probability, as shown in the inset of Figure 8. Here, <*P*_trans_> increases with *f* via the scaling law <*P*_trans_> ~ *f ^β^*, and the value of the exponent *β* shows two regions: *β* = 0.9 for the fast translocation region with a strong driving force (*f* > 3) and *β* = 1.6 for the slow translocation region with a weak driving force (*f* < 3). The translocation behavior of the hairpin polymer is different in these two regions. When *f* > 3, the translocation time <*τ*> and the translocation probability <*P*_trans_> of the hairpin polymer are close to those of the linear polymer and independent of the interaction *ε*_stem_. This is because the hairpin structure can be easily broken into a linear chain under a very strong driving force, so that the translocation process is only dependent on the driving force and not the hairpin structure. When *f* < 3, the translocation for a linear-like polymer and unsteady hairpin polymer is similar to that of the linear chain, while the steady hairpin polymer exhibits unique translocation behavior. As the driving force *f* decreases, the translocation time <*τ*> for the steady polymer initially increases rapidly and then decreases. It shows a non-monotonic dependence on *f*, and the maximum point of the translocation time occurs at around *f* = 0.4. The decrease in <*τ*> in the region of *f* < 0.4 is related to the rapid drop in the translocation probability <*P*_trans_>.

In order to further investigate the non-monotonic influence of the driving force on the displacement time, we enlarged the translocation time curve near *f* = 0.4 for the hairpin polymer with attractive strength *ε*_stem_ = 5, as shown in Figure 9, and the insets depict the free-energy landscape corresponding to different driving forces. From the free-energy landscape, we can intuitively understand the influence of the driving force on the translocation process. Once the hairpin structure and HB interaction are fixed, the height of the left barrier *H*_left_ mainly depends on the value of *f.* As shown in inset (b) of Figure 9, the translocation time reaches its maximum when *f* = 0.4, and the barrier height *H*_left_ = *H*_right_. The probability of the polymer moving forward and backward is the same, resulting in a long waiting time in the potential well. Meanwhile, for the condition of *f* > 0.4, which is shown in inset (c), *H*_left_ >> *H*_right_, and the monomers can pass through the nanopore easily. As the driving force facilitates the unfolding of the hairpin structure and the translocation process in this region, the translocation time <*τ*> decreases with increasing *f*. Meanwhile, when the driving force *f* is very weak (*f* < 0.4, shown in inset (a)), the free-energy barrier *H*_left_ << *H*_right_, and the depth of the energy well depends on *H*_left_, which decreases with decreasing *f*. Thus, the translocation time decreases in this region, and the decreasing *f* also causes an increase in *H*_right_. Consequently, fewer polymers with high free energy can pass through the nanopore, leading to a lower probability and short translocation time.

The influence of the length of the stem part *N*_stem_ on the translocation time <*τ*> is also studied, and the results are shown in Figure 10. The lengths of the overhang part and loop part are set as *N*_hang_ = 30 and *N*_loop_ = 5, respectively, and the length of the stem part *N*_stem_ varies from 0 to 10. The attractive strength *ε*_stem_ is set as 3 and 3.5 and the driving force is set as *f* = 0.5 and 6 for slow and fast translocation conditions, respectively. We can see that, when the driving force is large, <*τ*> is very close to <*τ*_0_> and increases linearly with *N*_stem_. Meanwhile, under the weak driving force condition, <*τ*> increases quickly with *N*_stem_ and thus separates from <*τ*_0_> at a large *N*_stem_. The dependence of <*τ*> on *N*_stem_ is nonlinear because the two strands in the stem part tend to distort into a steady helix structure to increase the entropy. Thus, the effect of the driving force in dissociating the base pairs during translocation can be divided into two components, with one pulling parallel to the bases to unzip the base pairs and one stretching transverse to the base pairs to unwind the helical stem part into two parallel strands [38]. The time scale for the unzipping process is proportional to the number of base pairs, while the unwinding process is much more complicated. The greater the number of base pairs, the more complicated the process for the helical stem part and the more time needed for the unwinding process.

To confirm this, we studied the translocation process for polymers with a gap in the stem part. As shown in the inset of Figure 10, there are 10 base pairs in the stem part of the polymer. The base pairs are marked as 1 to 10 sequentially with the first base pairs near the nanopore, and *i*_gap_ is defined as the *i*th base pair, which has no hydrogen bond. When 1 < *i*_gap_ < 10, the stem part can be separated into two stems: stem1 and stem2. The lengths of these two stems are *N*_stem1_ = *i*_gap_−1 and *N*_stem2_ = 9 − *N*_stem1_, respectively. The inset shows the relationship between the mean translocation time for the hairpin polymer with a gap in the stem part <*τ*_gap_> and the location of the gap *i*_gap_. The stem interaction is set as *ε*_stem_ = 3, and we find that <*τ*_gap_> is smaller than that for polymers with a subsequent 10 bps in the stem part, indicating that the existence of the gap weakens the relationship between the two stem parts, so that it takes less time to unwind the structure compared to the hairpin polymers with no gap. We also find that the translocation time <*τ*_gap_> is related to the location of the gap: when the gap is located at the middle of the stem part, the translocation time is shorter than that for polymers with a gap near the two ends of the stem part.

## 4. Conclusions

This paper presents a study of the translocation of polynucleotides with a hairpin structure through nanopores, using both Langevin dynamics simulations and the Fokker–Planck theory. The results show that there are three regions in the translocation process of hairpin polymers, and the translocation time and probability are related to the width (*m** ~ *N*_hang_ + *N*_stem_) and depth (minimum values in *H*_left_ and *H*_right_) of the energy well in the free-energy landscape. (1) When the driving force is very large or the interaction is small, *H*_left_ >> *H*_right_, and the width and depth of the energy well are both small, the monomers can pass through the nanopore easily, so the translocation process is close to that of the linear chain. (2) With an increase in the stem interaction and a decrease in the driving force, *H*_left_ decreases while *H*_right_ increases, and the width and depth of the well both increase. The monomers still tend to translocate into the *trans* side, but a longer translocation time is needed, corresponding to the rapidly increasing region of translocation. (3) Under the condition of a hairpin polymer with a steady structure and under a very weak driving force, *H*_left_ << *H*_right_, and the small depth of the well (*H*_left_) leads to a short translocation time. As the width of the energy well reaches the maximum (*m** = 0), the whole chain can thread back into the *cis* side, decreasing the possibility of the translocation of the hairpin polymer. The various translocation behaviors of the hairpin polymer indicate the possibility of hairpin structure detection during nanopore translocation.

These explanations seem reasonable under the assumption that the translocation of hairpin polymers can be treated as a normal diffusion process. Meanwhile, due to the HB interaction, the monomers in the stem part may be trapped near the nanopore for long periods of time and undergo a sub-diffusion process during translocation. The sub-diffusion of these monomers will have a certain impact on the accuracy of the LD simulation results. We minimized the impact of sub-diffusion on the entire translocation process by controlling the length and interaction strength of the stem part. For the further study of the situation in which sub-diffusion dominates the translocation process, simulation methods such as the fractional Brownian motion (FBM) and the fractional Langevin equation (FLE) for non-Brownian motion, as well as theoretical equations such as the fractional Fokker–Planck equation for continuous-time random walks, should be applied to ensure the accuracy of the results.

## Figures and Tables

**Figure 1 molecules-29-04042-f001:**
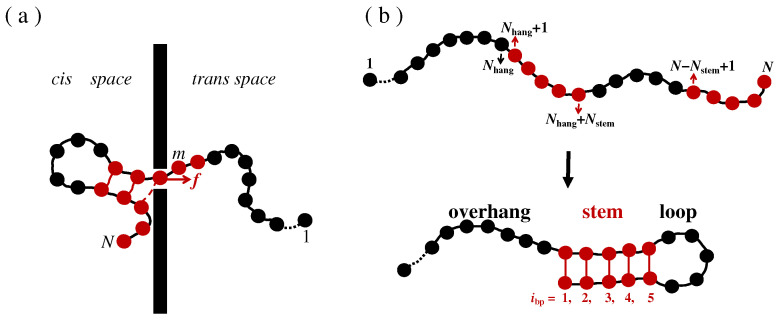
(**a**) A 2D sketch of the translocation of a hairpin polymer through a nanopore; (**b**) the coarse-grained bead-spring model for a linear chain (up) folding into a hairpin structure (down).

**Figure 2 molecules-29-04042-f002:**
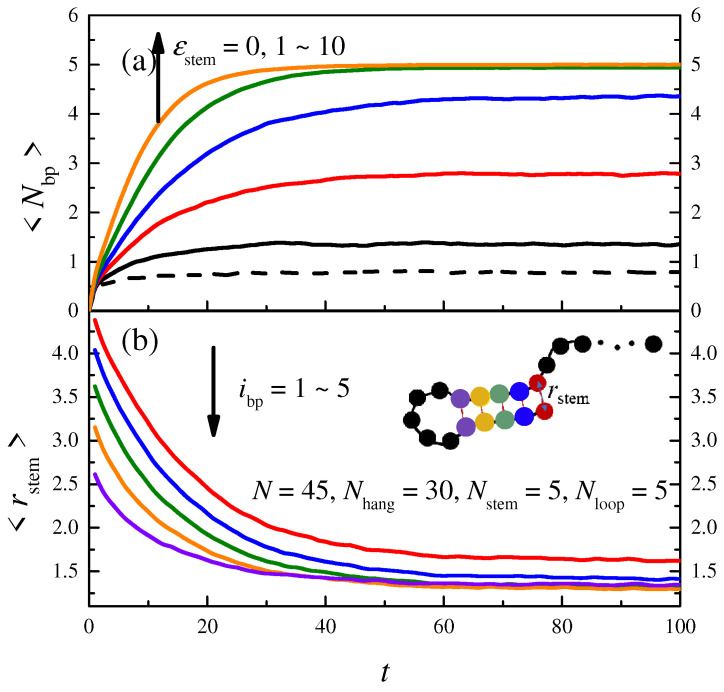
The plot (**b**) shows the variation in the mean distance in bps <*r*_stem_> with simulation time *t*; the interaction strength for the HB potential is set as *ε*_stem_ = 6. The plot (**a**) shows the dependence of the number of hydrogen bonds <*N*_bp_> on *t*; the lines from bottom to top represent polymers with attractive strength *ε*_stem_ = 1, 2, 3, 5, and 10; the dashed line represents the linear polymer chain.

**Figure 3 molecules-29-04042-f003:**
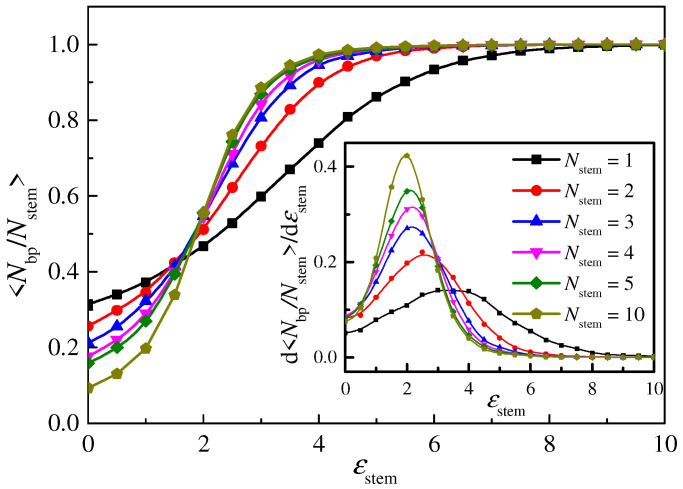
The dependence of the ratio <*N*_bp_/*N*_stem_> when *t* = 100 on the interaction *ε*_stem_ for polymers with different numbers of base pairs *N*_stem_. The lines from bottom to top represent polymers with a base pair number *N*_stem_ = 1~10. The inset shows the changing rate d<*N*_bp_/*N*_stem_>/d*ε*_stem_ in dependence on *ε*_stem_. The lengths of the overhang and loop parts are set as *N*_hang_ = 30 and *N*_loop_ = 5, respectively.

**Figure 4 molecules-29-04042-f004:**
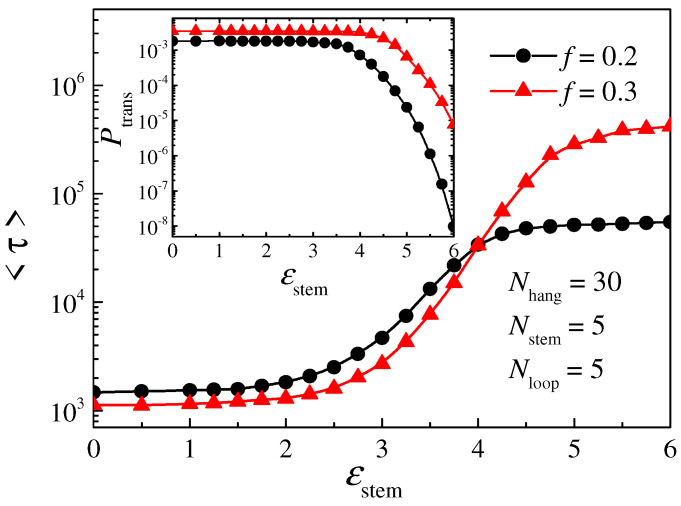
The dependence of the translocation time <*τ*> and the translocation probability <*P*_trans_> (the inset) on the attractive strength *ε*_stem_. Different lines represent polymers under different driving forces; the length of each part of the polymer chain is *N*_hang_ = 30, *N*_stem_ = 5, *N*_loop_ = 5, respectively.

**Figure 5 molecules-29-04042-f005:**
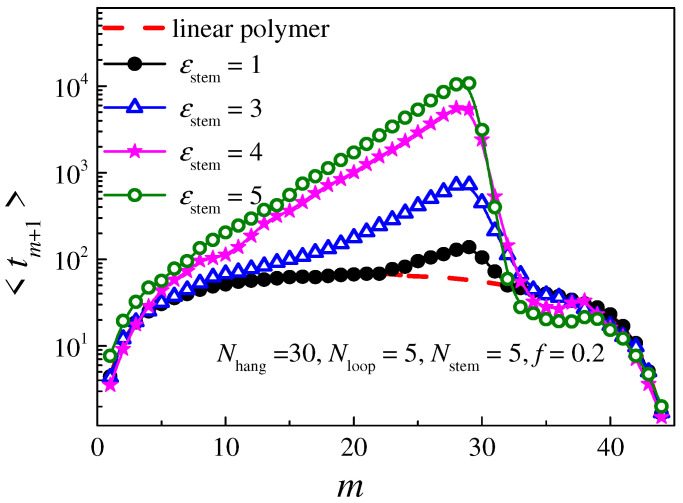
The distribution of the mean residence time <*t_m_*_+1_> for the (*m*+1)th monomer remaining at the nanopore. The length of each part of the hairpin polymer is set as *N*_hang_ = 30, *N*_stem_ = 5, and *N*_loop_ = 5, respectively; the driving force is set as *f* = 0.2. The red dashed line represents the linear polymer with length *N* = 45; the other lines represent hairpin polymers with different attractive strength *ε*_stem_.

**Figure 6 molecules-29-04042-f006:**
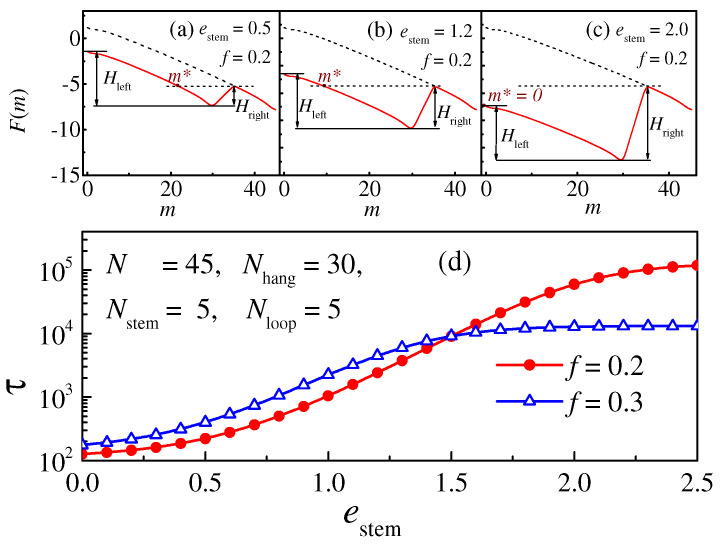
Plots (**a**–**c**) display the free-energy landscape during translocation for polymers with different stem interactions *e*_stem_; the length of each part of the hairpin polymer is set as *N* = 45, *N*_hang_ = 30, *N*_stem_ = 5, and *N*_loop_ = 5, respectively, and the driving force is set as *f* = 0.2. The dashed line represents the linear polymer with contour length *N* = 45. Plot (**d**) illustrates the translocation time for the absorbing–absorbing boundary condition obtained from the Fokker–Planck equation, which varies with *e*_stem_.

**Figure 7 molecules-29-04042-f007:**
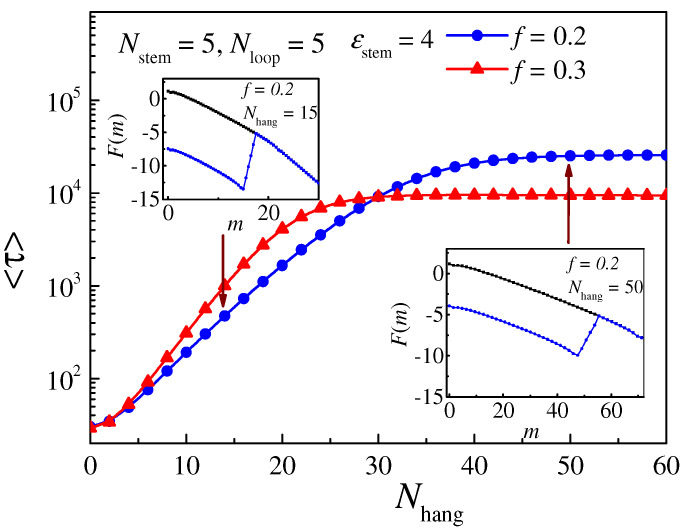
The relationship between the translocation time *τ* and the length of the overhang part *N*_hang_. The lengths of the stem and loop part are set as *N*_stem_ = 5 and *N*_loop_ = 5, respectively; the length of the overhang part varies from 0 to 60; and the driving force is set as *f* = 0.2 (blue line with circle point) and *f* = 0.3 (red line with triangle point), respectively. The attractive strength *ε*_stem_ = 4. The insets show the free-energy landscape for the hairpin polymer in two regions.

**Figure 8 molecules-29-04042-f008:**
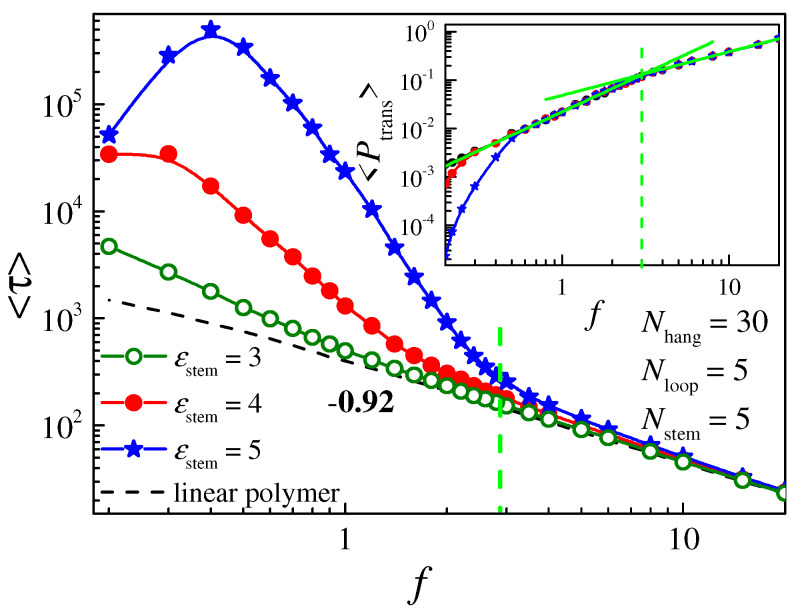
The relationship between the translocation time <*τ*> and the driving force *f*. The inset shows the dependence of the translocation probability Ptrans on *f*. The dashed line represents the translocation time for a linear polymer with a polymer length *N* = 45; the solid lines represent the translocation time for hairpin polymers with different attractive interactions; the lengths of each part of the hairpin polymer are set as *N* = 45, *N*_hang_ = 30, *N*_stem_ = 5, and *N*_loop_ = 5, respectively.

**Figure 9 molecules-29-04042-f009:**
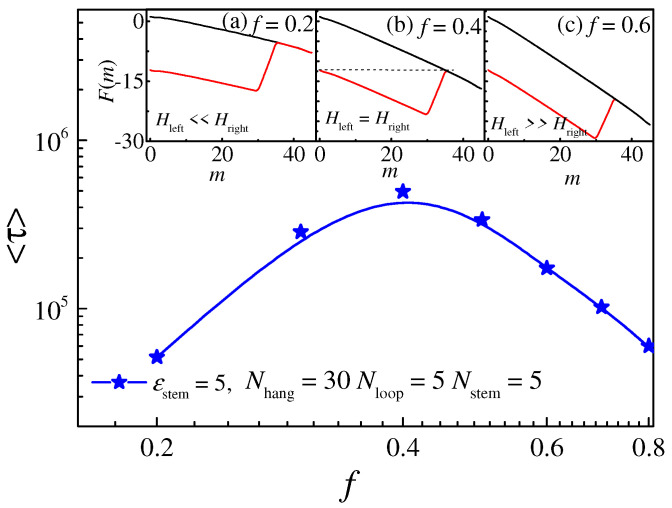
The relationship between the translocation time <*τ*> and the driving force *f* for polymers with interaction strength *ε*_stem_ = 5. Red lines in the insets (**a**−**c**) show the free-energy landscapes for hairpin polymer with *f* = 0.2, 0.4, and 0.6, respectively, black lines represent the free-energy landscape for the linear polymer. The lengths of each part of the hairpin polymer are set as *N* = 45, *N*_hang_ = 30, *N*_stem_ = 5, and *N*_loop_ = 5, respectively.

**Figure 10 molecules-29-04042-f010:**
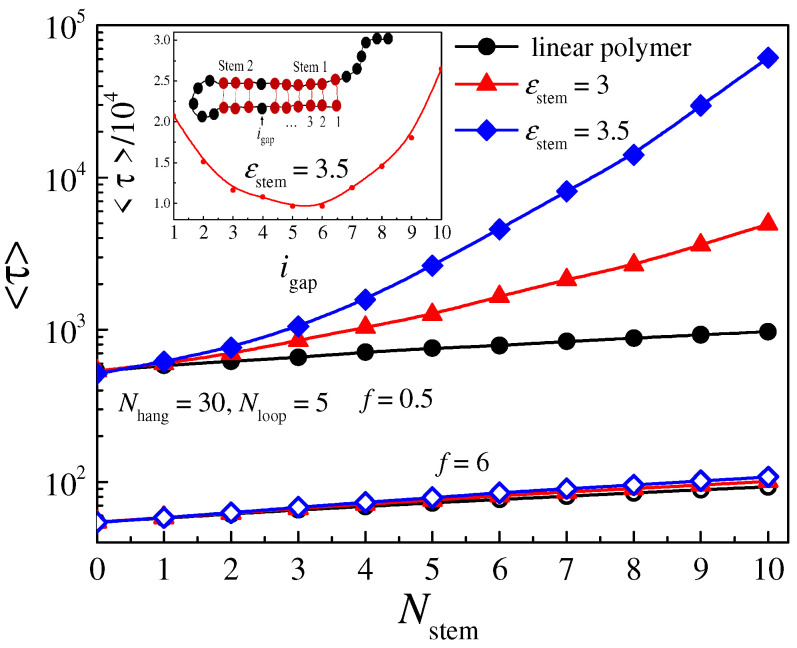
The relationship between the translocation time <*τ*> and the length of the stem part *N*_stem_. Solid symbols represent weak driving conditions, and open symbols represent strong driving conditions. The inset shows the dependence of the translocation time for polymers with a gap in the stem part <*τ*_gap_> on the gap location *i*_gap_.

## Data Availability

The original contributions presented in the study are included in the article, further inquiries can be directed to the corresponding authors.

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
