# Peer review of "Langevin Dynamics Study on the Driven Translocation of Polymer Chains with a Hairpin Structure"

_molecules, 2024, doi:10.3390/molecules29174042_

Round 1

Reviewer 1 Report

Comments and Suggestions for Authors

In the article entitled: Langevin dynamics study on the driven translocation of polymer chains with hairpin structure authors discuss the theoretical methodology to predict the spatial structure. Take into consideration the problem of macromolecules with high numbers of atoms, from one hand and the requested time of molecular dynamics from the second one. Both of the above reasons make the dynamics calculation highly CPU time-consuming. The authors correctly implied the Fokker-Planck theory with Leonard-Jones potential. The above make, presented by them, Langevin dynamics attractive for future consideration. The article is well-written and readable, moreover, the materials and method parts clearly present the author's strategy. The above was complemented by correctly cited references. From the editorial point, I have found cosmetic mistakes like the lack of time units in Figure 2.

Therefore, taking the above into consideration make the article interesting and worth publication.

However, authors should discuss the fact that DNA is the polyanion what makes it different from many other polymers. Therefore, the authors in their deliberation, the counterion should be taken into consideration too. I hope that the authors will be able to provide a suitable answer to my question.

Reviewer 2 Report

Comments and Suggestions for Authors

This report provides a comprehensive analysis of the manuscript titled "Langevin dynamics study on the driven translocation of poly-2mer chains with hairpin structure" authored by Fan Wu, Xiao Yang, Chao Wang, Bin Zhao, and Meng-Bo Luo.

The authors investigate the translocation process of a model polymer chain with a hairpin structure using Langevin dynamics simulations in conjunction with a random walk model based on the free energy landscape. Their study demonstrates that the dynamics of hairpin polymer translocation through a nanopore are influenced by the hairpin structure, as revealed through numerical simulations. Notably, the translocation time is observed to increase when the structure deviates from a linear polymer chain.

The study's motivation is clearly defined and holds relevance in the realms of biological and chemical transport. The analysis is robust and is substantiated by numerical simulations. In summary, I support the publication based on my overall positive impression of the work.

Minor comments on the manuscript:

1. The phrase "The translocation time unsteady hairpin" lacks clarity and should be revised for better understanding.

2. The manuscript could benefit from discussing the connection between the translocation dynamics of a polymer and concepts such as fractional Brownian motion (FBM) or the fractional generalized Langevin equation, providing insights for further exploration.

3. Further elaboration on noise in the polymer dynamics, including the fluctuation-dissipation relation linking noise intensity (temperature) and viscosity, would enhance the explanation.

4. Providing additional details on the derivation of Eq. (4) and the explicit form of the diffusion coefficient $D(m)$, along with addressing whether the diffusion coefficient depends on $m$, would be beneficial for readers' comprehension.

5. Clarification regarding the arrows in Fig. 6(d) is needed for better interpretation of the figure.

6. Figure 8 presents intriguing results, particularly the non-trivial peak for $\varepsilon_{\rm stem}=5$. 

Including discussions on this aspect would enhance the manuscript's impact and relevance.
